# Iron and DHA in Infant Formula Purchased in the US Fails to Meet European Nutrition Requirements

**DOI:** 10.3390/nu15081812

**Published:** 2023-04-08

**Authors:** Alexander Strzalkowski, Grace Black, Bridget E. Young

**Affiliations:** 1Department of Pediatrics, Complex Care Service, Boston Children’s Hospital, Boston, MA 02115, USA; 2Department of Pediatrics, University of Rochester School of Medicine and Dentistry, Rochester, NY 14642, USA; 3Department of Pediatrics, Breastfeeding and Lactation Medicine, University of Rochester School of Medicine and Dentistry, Rochester, NY 14642, USA

**Keywords:** infant formula, human milk, iron, docosahexaenoic acid

## Abstract

Requirements for iron and docosahexaenoic acid (DHA) content of infant formula varies by country. Powdered full-term infant formula purchase data from all major physical stores in the US between 2017–2019 were obtained from CIRCANA, Inc. Iron and DHA composition and scoop sizes for each formula were obtained from manufacturers. The equivalent liquid ounces of prepared formula were calculated. Average iron and DHA content were compared between formula types and to both US and European formula composition requirements. These data represent 55.8 billion ounces of formula. The average iron content of all formula purchased was: 1.80 mg/100 kcal. This iron concentration is within the FDA regulations. However, it exceeds the maximum allowable iron concentration of infant formula (Stage 1) set by the European Commission of 1.3 mg/100 kcal. A total of 96% of formula purchased had an iron concentration of >1.3 mg/100 kcal. DHA is not a required ingredient in US formulas. The average DHA content of all formula purchased was: 12.6 mg/100 kcal. This DHA concentration is far below the minimum required DHA concentrations of infant formula (Stage 1) and follow-on formula (Stage 2) set by the European Commission of 20 mg/100 kcal. These are novel insights into the iron and DHA intake of formula-fed infants in the US. As international infant formulas have entered the US market due to the formula shortage, parents and providers need to be aware of regulatory differences in formula nutrient composition.

## 1. Introduction

Exclusive breastfeeding is universally recommended [1,2]. However, for numerous reasons, approximately 75% of US infants are fed infant formula before the age of 6 months [3]. Infant formula is a human milk substitute that safely meets the nutrient requirements of infants, if fed exclusively between 0–6 months and when combined with complementary foods after 6 months.

In the United States, the Food and Drug Administration (FDA) regulates what ingredients can be included in infant formula, and the allowable minimum and maximum concentrations of 28 essential nutrients [4]. Internationally, alternative analogous regulatory bodies serve this role. For example, the European Commission sets ingredient regulations and nutrient requirements for infant formula sold in the European Union [5].

CODEX ALIMENTARIUS is an international food standards organization (supported by the Food and Agriculture Organization of the United Nations and the World Health Organization) which provides consensus guidelines for many nutrient components of infant formula, including both “stage 1” infant formula (for infants 0–6 months) [6], and “stage 2” or “follow on formula” (for infants >6 months) [7]. Many countries adhere to CODEX guidelines for formula regulation, including the United Kingdom, those in the European Union, Australia, and New Zealand. The US does not. This results in different regulations for infant formula between countries. Particular differences include staging: these other countries manufacture formulas that adhere to different nutrient guidelines for Stage 1 vs. Stage 2 formulas, whereas in the US, any “infant formula” can be fed to an infant from birth to 12 months. Maximum allowable iron concentrations differ between countries. Lastly, docosahexaenoic acid is a required ingredient in infant formula sold in the European Union, whereas it is not required in US formulas.

In the past decade, illegally imported foreign formulas, particularly from Europe, have grown in popularity in the US [8]. Additionally, the US infant formula shortage (2022–2023) resulted in mass importation of foreign formulas. As a result, in May of 2022, the FDA released guidance for industry allowing foreign formulas to apply for temporary approval and importation into the US in order to help ease the infant formula shortage [9]. In January 2023, the FDA announced that several foreign formulas had taken the appropriate steps to remain permanently available in the US [10].

For many months, infant formula manufactured under different regulatory guidelines have been available to US parents, resulting in significant confusion [11]. In this context, we sought to compare the iron and DHA concentrations of formula purchased in the US prior to COVID-19 and the infant formula shortage with the regulatory guidelines utilized internationally and reflected in the new foreign products now legally available to US parents.

## 2. Materials and Methods

### 2.1. Data Acquisition

National level US purchasing data of all powdered infant formula purchased from major physical stores (excluding Costco, which did not provide data) from 2017 through 2019 were acquired from CIRCANA, Inc., Chicago, IL, USA. Purchases of ready-to-feed and concentrated formula products were not included. This time period was chosen to capture normal purchasing habits prior to the COVID-19 pandemic and the infant formula shortage. These data included market average dollar sales and weight sold of individual formula products in 2017, 2018, and 2019 that sold for >USD 7000 each year. The iron and DHA content of each formula and scoop size (in grams) was obtained from the manufacturer for each infant powdered infant formula. “Formula equivalent ounces consumed” were calculated based on the total weight sold and the individual formula scoop size, assuming a volume expansion of 1.1 ounce of prepared formula per ounce of water. These analyses are performed under the assumption that all formula purchased was prepared as directed.

### 2.2. Formula Categorization

Purchases of premature and toddler formulations were not included. Protein hydrolyzation status was classified as intact, partially hydrolyzed, fully hydrolyzed, or amino acid-based. Hypoallergenic formulas were those that had a protein source of either fully hydrolyzed or amino acid-based formula.

### 2.3. Regulatory Guidelines

Average iron and DHA concentrations of formulas purchased in the US were compared to the limits set by the following regulatory bodies:The Food and Drug Administration (FDA) sets nutrient requirements for infant formula (infants 0–12 months) produced and marketed in the US [4].The European Commission sets nutrient requirements for infant formula (infants 0–6 months) and follow-on formula (infants 6–12 months) produced and marketed in the European Union [5].CODEX Alimentarius (representative of the Food and Agriculture Organization of the United Nations and the World Health Organization) proposes suggested nutrient requirements for infant and follow-on formula that nations can adopt individually [6,7].

A summary of these nutrient regulations for both iron and DHA are given in Table 1.

### 2.4. Statistical Analysis

Average micronutrient concentrations reported were calculated as the weighted mean of product concentrations, weighted by amount of product purchased (in equivalent liquid ounces). All statistical analyses were performed using JMP (SAS Institute, Cary, NC, USA).

## 3. Results

These data represent USD 8.1 billion of sales over this three-year period, equating to 55.8 billion ounces of formula purchased. The macronutrient composition of formula purchased has recently been published [12]. In summary, 5.6% of formula purchased was soy-based. Totals of 64.1% and 24.8% of formula purchased were intact-protein and partially hydrolyzed protein dairy formulas, respectively. A total of 5.5% of formula purchased was hypoallergenic (Figure 1).

Powdered infant formula purchased from all major physical retailers in the US (excluding Costco) between 2017–2019, broken down by formula protein type and expressed as percentage of all formula purchased (by volume).

### 3.1. Iron Content of Formula Purchased

The average iron content of all formula purchased was 1.80 mg/100 kcal. The maximum concentration of iron in any product purchased was 1.9 mg/100 kcal. The lowest concentration of iron in any product purchased was 1.0 mg/100 kcal. There was slight variation in the average iron content of formula based on hydrolyzation status, with intact protein formulas representing the highest average iron concentrations (1.85 mg/100 kcal) and partially hydrolyzed dairy-based formulas representing the lowest average iron concentrations (1.67 mg/100 kcal) (Figure 2).

The overall average iron concentration (1.80 mg/100 kcal) is within the FDA regulations. However, it exceeds the maximum allowable iron concentration of infant formula (Stage 1) set by the European Commission at 1.3 mg/100 kcal. A total of 96% of formula purchased had an iron concentration of >1.3 mg/100 kcal.

### 3.2. DHA Content of Formula Purchased

The average DHA content of all formula purchased was 12.6 mg/100 kcal. The maximum concentration of DHA in any product purchased was 17 mg/100 kcal. Only 0.001% of formula purchased was not supplemented with DHA and ARA. There was slight variation in the average DHA content of formula based on hydrolyzation status with partially hydrolyzed dairy-based protein formulas representing the highest average DHA concentrations (15.6 mg/100 kcal) and amino acid formulas representing the lowest average DHA concentrations (9.0 mg/100 kcal) (Figure 3).

The overall average DHA concentration (12.6 mg/100 kcal) is far below the minimum required DHA concentrations of infant formula (Stage 1) and follow-on formula (Stage 2) set by the European Commission of 20 mg/100 kcal.

## 4. Discussion

To our knowledge, this is the first description of iron and DHA concentrations of formula purchased in the United States on a national level. We demonstrate that the average iron and DHA concentrations of purchased formula fall above and below the requirements for formulas in the EU, respectively. This is particularly relevant given the current influx of European and other foreign formulas into the United States market due to the infant formula shortage.

Between 2017–2019, 96% of infant formula purchased contained iron content above the maximum allowable iron concentrations for European Stage 1 formula. Our data do not provide resolution into the age of the infant for which formula was purchased. Regardless of the proportion of infants represented by these data that are <6 months of age, the majority of these young infants are exposed to “toxic” levels of iron, according to standards set by the European Commission, whereas this iron exposure is acceptable according to the standards set by the FDA.

The average iron concentration of formula purchased in the US was 1.80 mg/100 kcal. Iron concentration in human milk is significantly lower, with an approximate average of 0.03–0.05 mg/100 mL (which is equivalent to: 0.009–0.015 mg/100 kcal, assuming human milk is 68 kcal/100 mL) [13,14]. The bioavailability of iron from dairy- and soy-based infant formula is roughly 40% and 84% lower than that of human milk [14], and thus higher concentrations of iron are required to meet the iron needs of exclusively formula-fed infants.

Studies of low-birth-weight infants suggest that feeding infant formula with 3.04 mg/100 kcal does not improve growth, iron status, or development more than a formula with 1.97 mg/100 kcal, and may result in compromised zinc and copper absorption as well as more oxidative stress [15]. Previous randomized controlled trials of healthy term infants have randomized infants to receive either high-iron formula (1.87 mg/100 kcal), low-iron formula (0.34 mg/100 kcal) or breast milk or cow’s milk with no supplemental iron from 6–12 months of age. This trial established that zero iron supplementation resulted in increased anemia and delays in crawling and processing information, as well as decreases in positive affect [16]. Between the low- vs. high-iron formula, there was no difference in the prevalence of anemia, but the high iron group did have higher hemoglobin and ferritin values at 12 months [17]. However, when followed up at ~16 years of age, the adolescents who had received the high-iron formula between 6–12 months of life exhibited worse visual memory, worse arithmetic achievement, and worse reading comprehension compared to the low-iron group [18]. In this study, adolescents in the low-iron group exhibited worse visual motor integration than the high-iron group only if they also had low hemoglobin during infancy [18]. It is important to note that the low-iron group in this study was only provided 0.34 mg/100 kcal iron. This is well below the level at which the FDA requires a label of “supplemental iron may be necessary” to be added to a product (when iron <1.0 mg/100 kcal). No standard infant formulas in the US have an iron concentration below 1.0 mg/100 kcal. Given the detrimental results observed among adolescents who consumed infant formula with 1.87 mg/100 kcal iron between 6–12 months of age, it is potentially concerning that the average iron concentration in US formulas purchased during 2017–2019 was 1.80 mg/100 kcal (and higher, at 1.85 mg/100 kcal in formula with an intact protein source).

No studies have assessed the long term developmental outcomes of exclusively feeding healthy term infants formula with iron concentrations of 1.80 mg/100 kcal. However, the studies above suggest that lower iron concentrations are likely enough to ensure sufficient iron status without causing adverse outcomes. This reasoning is reflected in the guidelines set by the European Commission and by the European formulas now available permanently in the US. Regardless of the difference in regulatory bodies, it is crucial that pediatricians recognize that there are many infant formulas now available in the US that have significantly lower iron concentrations than have previously been available to US infants.

DHA concentrations in human milk range significantly [19] based on the maternal dietary intake of DHA [20]. Average concentration is roughly 0.32% of total fat (which is 14.1 mg/100 kcal, assuming human milk has 3.0 g fat/100 mL [20] and 68 kcal/100 mL).

The scientific literature relating to the benefits of long-chain polyunsaturated fatty acids (LC-PUFAs) supplementation, particularly DHA, in the term infant is vast and often conflicting. Meta-analyses in 2008 concluded that there may be some benefit to omega-3 fatty acid supplementation on visual acuity and cognition, but this was not consistently detected [21]. It was noted that in instances of DHA supplementation, ARA should also be included [21]. A Cochrane review on the topic in 2011 concluded that the beneficial effects of DHA supplementation on visual acuity among full-term infants had not been demonstrated consistently enough to warrant the routine supplementation of term infant formula with LCPUFA [22]. In 2012, another meta-analysis revealed that there was no beneficial impact of LC-PUFA supplementation in the first 12 months of life on infant cognition, as assessed by Bailey scores [23]. Additionally, in 2012, a double-masked randomized control trial (*n* = 141) of various concentrations of DHA in infant formula (0%, 0.32%, 0.64%, or 0.96% fatty acids with 0.64% fatty acids as ARA) over the first 12 months found no impact on the various cognitive assessments of school readiness and vocabulary [24]. However, a similar blinded randomized control trial (*n* = 81) of the same levels of DHA supplementation in infant formula over the first 12 months showed that higher levels of DHA supplementation did result in increased scores on some but not all cognitive testing, including: increased rule-learning and inhibition tasks at age 3, increased Peabody picture vocabulary test scores at age 5, and Weschler primary preschool scales of intelligence at age 6 [25]. There were no effects of LCPUFA on spatial memory or simple inhibition to advanced problem-solving tasks [25]. Following these trials, in 2015 the European Commission updated guidelines to require supplemental DHA at between 20–50 mg/100 kcal in all infant and follow-on formula by the year 2020 [5]. This updated regulation did not include a requirement for the co-addition of ARA.

It has long been recommended that, if DHA is added to infant formula, ARA also be added to prevent deleterious impacts on growth [21]. This is reflected in the FDA regulations for infant formula nutrient requirements: while DHA is not required, if it is added, then ARA needs to be included at a ratio with DHA between 1:1–1:2 (Table 1) [4]. A recent consensus statement of the European Academy of Peadiatrics and the Child Health Foundation stated that DHA should be required in infant and follow-on formula between 0.3–0.5% of total fatty acids, but that ARA should also be required in tandem at concentrations at least equal to that of DHA [26]. These are the first data to suggest that, prior to 2020, US formula-fed infants were consuming 12.6 mg/100 kcal DHA on average. In the current context of the infant formula shortage, the European formulas now available all have ≥20 mg/100 kcal of DHA. DHA is a popular source of marketing messaging on formula cans and websites in the US [27]. The conflict between regulatory requirements for DHA concentrations between both sets of formula currently available to US parents may be a source of confusion for both parents and providers.

As of January 2023, it is apparent that many international formula brands will be remaining in the US on a permanent basis [10], highlighting the need to ensure clinical providers are aware of differences in the composition and regulation of nutrients such as iron and DHA. It also serves as a point of advocacy for the consensus regulations of infant formula composition in a global economy.

This study has many strengths. It represents the largest description of infant formula purchasing habits conducted in the US. It does not include purchases of liquid formula. However, as iron and DHA concentrations do not differ between powder vs. liquid formulations of products in the US, this weakness is not expected to impact results. Our data are not connected with any infant characteristics or outcomes, which are necessary to estimate the nutritional needs of individual infants and compare formula purchased to the appropriate Stage 1 vs. Stage 2 formula regulations.

## 5. Conclusions

In summary, these data highlight that formula purchased in the US between 2017–2019 had higher iron and lower DHA compared to foreign formulas, particularly those manufactured in the EU. This difference is clinically relevant in the US now that many international formula brands have entered and will remain available on the US market as a result of the 2022–2023 infant formula shortage.

## Figures and Tables

**Figure 1 nutrients-15-01812-f001:**
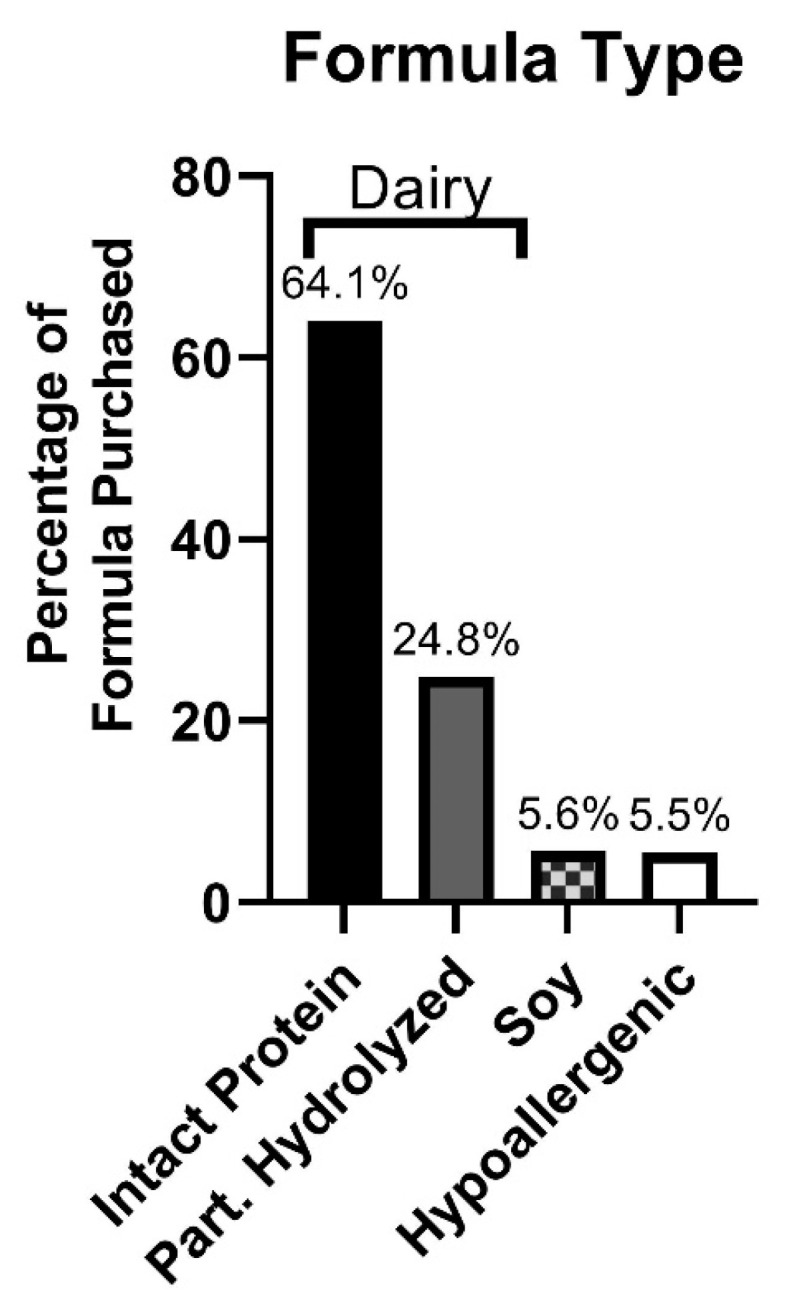
Types of infant formula purchased in the US between 2017–2019.

**Figure 2 nutrients-15-01812-f002:**
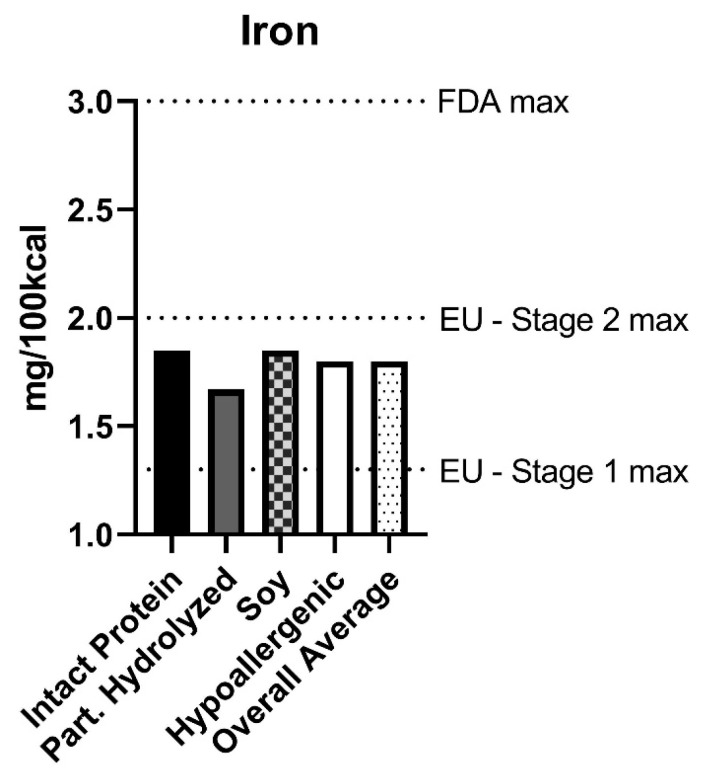
Iron content of infant formula purchased in the US between 2017–2019. Powdered infant formula purchased from all major physical retailers (excluding Costco) in the US between 2017–2019. Average iron content (mg/100 kcal), controlling for volume purchased, is reported in each category of formula. Horizontal lines represent the European Union (EU) maximum allowable iron concentration for Stage 1 and 2 formula and the Food and Drug Administration (FDA) maximum allowable iron concentration for all US infant formula.

**Figure 3 nutrients-15-01812-f003:**
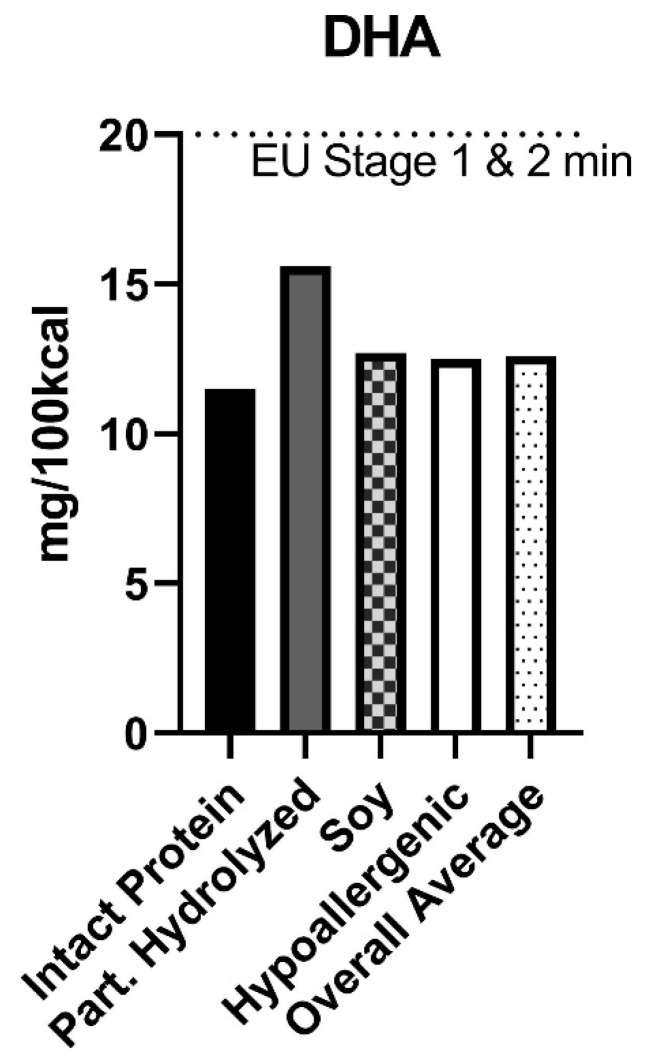
DHA content of infant formula purchased in the US between 2017–2019. Powdered infant formula purchased from all major physical retailers (excluding Costco) in the US between 2017–2019. The average docosahexaenoic acid (DHA) content (mg/100 kcal), controlling for volume purchased, is reported in each category of formula. The horizontal line represents the European Union (EU) minimum required concentration for both Stage 1 and 2 formulas. DHA is not a required ingredient in US formulas.

**Table 1 nutrients-15-01812-t001:** International regulatory bodies requirements for iron and DHA in infant formula.

Regulatory Body	Age (Months)	Formula Type	Minimum	Maximum
**Iron (mg/100 kcal)**				
Food and Drug Administration	0–12	All	0.15 ^1^	3.0
European Commission	0–6	Non-Soy-Based	0.3	1.3
	6–12	Non-Soy-Based	0.6	2.0
	0–6	Soy-Based	0.45	2.0
	6–12	Soy-Based	0.9	2.5
CODEX Alimentarius	0–6	All	0.45	-
	6–12	All	1.0	2.0
**Docosahexaenoic Acid (DHA) (mg/100 kcal)**				
Food and Drug Administration ^2^	All	All	Not Required	
European Commission ^3^	0–6	All	20	50
	6–12	All	20	50
CODEX Alimentarius ^4^	0–6	All	Not Required	
	6–12	All	Not Required	

^1^ Formulas containing less than 1.0 mg/100 kcal must carry the phrase “Additional Iron may be necessary” on the front product packaging. ^2^ DHA is not a required nutrient in US formulas. When it is included, then the arachidonic acid (ARA) to DHA ratio must be between 1:1 to 2:1. ^3^ Eicosatetraenoic acid (EPA) concentrations do not exceed concentrations of DHA. ^4^ DHA requirements are not suggested by CODEX Alimentarius. Yet, when DHA is included, it is recommended that ARA also be required at a minimum of the same concentration as DHA and that EPA concentrations not exceed concentrations of DHA.

## Data Availability

The data that support the findings of this study were secured from CIRCANA, Inc. Restrictions apply to the availability of these data. Requests for such data should be presented to CIRCANA, Inc.

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
