# Peer review of "Iron and DHA in Infant Formula Purchased in the US Fails to Meet European Nutrition Requirements"

_nutrients, 2023, doi:10.3390/nu15081812_

Round 1

Reviewer 1 Report

Manuscript ID: nutrients-2274641

TITLE: “Iron and DHA in infant formula purchased in the US fails to 2 meet European nutrition requirements” by Strzalkowski, Black and Young.

In this article, the authors made a comparison on the  iron and DHA concentrations between infant formula purchased in the US and other foreign formulas, particularly those manufactured in the EU. This work is interesting and well written, I suggest publication after minor revisions which are listed below.

Introduction.

Line 55-60. The authors stated that “….. the US infant formula shortage (2022-2023) resulted in mass importation of foreign formulas”. It would be interesting for the reader to know what caused the US infant formula shortage in 2022-2023. Was it because of the COVID-19 pandemic?

Material and Methods

Line 73. The authors should explain why have chosen data of powdered infant formula prior to the COVID-19 pandemic.

My best regards

Author Response

We wish to thank the reviewer for the kind words, and for the insightful comments. We have edited the manuscript to address these issues raised and believe the changes have improved the presentation.

TITLE: “Iron and DHA in infant formula purchased in the US fails to2 meet European nutrition requirements” by Strzalkowski, Black and Young. In this article, the authors made a comparison on the iron and DHA concentrations between infant formula purchased in the US and other foreign formulas, particularly those manufactured in the EU. This work is interesting and well written, I suggest publication after minor revisions which are listed below.

Introduction.

Line 55-60. The authors stated that “….. the US infant formula shortage (2022-2023) resulted in mass importation of foreign formulas”. It would be interesting for the reader to know what caused the US infant formula shortage in 2022-2023. Was it because of the COVID-19 pandemic?

Thank you for this suggestion. We have added text to this paragraph to explain that the shortage was partially due COVID-19 supply chain limitations. But in February 2022, the largest infant formula recall in US history was initiated (by Abbott) and included shutting down one of the main infant formula manufacturing plants in the country.  This was the main reason behind the ongoing US infant formula shortage.

Material and Methods

Line 73. The authors should explain why have chosen data of powdered infant formula prior to the COVID-19 pandemic.

Thank you.  We have added detail to this section to explain that purchases of ready-to-feed and concentrated formula products were not available in the database and thus not included. This particular time period was chosen to capture normal caregiver purchasing habits prior to the COVID-19 pandemic and the panic-buying that ensued in the US.  We also did not want to include purchase data from during the shortage as there were significantly limited formula options and availability during that time.

Reviewer 2 Report

General comments:

The problem (formula shortage and the presence of foreign formula in the US) and the objective of the work is well presented. the paper responds well to the presented problem. 

Major comments:

The paper answers the main question well, however it would gain in interest and novelty if the authors expanded more in the discussion on the impact of this change in purchasing behavior on nutrient intake versus recommendations. For pediatricians, knowing that nutrient content of European formulas differ from the US formulas is important but the impact on nutrient intake and health is equally as important. Would the authors be able to comment on, based on % purchasing of European formulas in 2022 (if available), the potential beneficial or detrimental impact of this change in purchasing behavior?

The Introduction explains that since 2022, a mass importation of European formulas was allowed into the US and that in 2023, "the FDA announced that several foreign formulas had taken the appropriate 60 steps to remain in the US permanently" It would be useful to clarify whether these steps included regulatory adaptation of their nutrient content (If the nutrient content needed to be adapted to import permanently into the US, the rationale for this work is lost). 

Is the average  iron and DHA  content of formulas bought in 2017- 2019 representative of the US formulas only (because European was illegally imported and thus not captured) OR did it also include some European formulae? Maybe clarify this in the text. 

Minor comments:

The authors mention that the scoop size was collected to calculate formula consumed. Was the amount of formula consumed used in this work? and if yes, how? it would be useful to report these results in the paper. If this was not used then remove from the methods.

The paragraph on formula staging regulations could be referenced (line 48 to 54). 

Author Response

We wish to thank the reviewers for the kind words, and for the insightful comments. We have edited the manuscript to address these issues raised and believe the changes have improved the presentation.

General comments:

The problem (formula shortage and the presence of foreign formula in the US) and the objective of the work is well presented. the paper responds well to the presented problem.

Major comments:

The paper answers the main question well, however it would gain in interest and novelty if the authors expanded more in the discussion on the impact of this change in purchasing behavior on nutrient intake versus recommendations. For pediatricians, knowing that nutrient content of European formulas differ from the US formulas is important but the impact on nutrient intake and health is equally as important.

This is a great point.  Unfortunately, we do not have data on actual intake, just purchases. However, we provide expanded text in the discussion to calculate what the representative difference in actual daily iron intake would be comparing the average US formula iron content vs popular European concentrations.  In short, exclusively formula fed infants < 6mo consuming a European vs US formula will have a 42-56% lower daily iron intake.  This is now included in the discussion, and we agree strengthens the paper.

Would the authors be able to comment on, based on % purchasing of European formulas in 2022 (if available), the potential beneficial or detrimental impact of this change in purchasing behavior?

We agree this would be valuable to know. Unfortunately, we don’t have that data and it is not publicly available in the US (unlike other countries). Thus, there is no place we can obtain it at the moment.

The Introduction explains that since 2022, a mass importation of European formulas was allowed into the US and that in 2023, "the FDA announced that several foreign formulas had taken the appropriate steps to remain in the US permanently" It would be useful to clarify whether these steps included regulatory adaptation of their nutrient content (If the nutrient content needed to be adapted to import permanently into the US, the rationale for this work is lost).

This is also a great point. The imported formulas did need to change any formulations that did not meet FDA requirements before initial import.  However, since the average iron content is well below the maximum and well above the minimum FDA requirements, this component of foreign formulas did not need to be changed.  Formulas with iron concentrations below 1.0mg/100kcal did need to add the phrase “additional iron may be necessary” to the front of the packaging.

Similarly, as there is no DHA requirement for formula in the US, the higher concentrations of DHA in European formulas is in line with FDA regulations (which do approve DHA as a potential optional ingredient for infant formula). We have added text to the discussion to clarify this point, as it is very important to context.

Is the average iron and DHA content of formulas bought in 2017- 2019 representative of the US formulas only (because European was illegally imported and thus not captured) OR did it also include some European formulae? Maybe clarify this in the text.

Thank you – this is another important clarification. The US data does not include any European formula purchases as European infant formula purchased during that time would have been all illegally obtained online via third parties and thus not available for purchase at established physical stores. We have added text to the methods to make this point clear.

Minor comments:

The authors mention that the scoop size was collected to calculate formula consumed. Was the amount of formula consumed used in this work? and if yes, how? it would be useful to report these results in the paper. If this was not used then remove from the methods.

Thank you for bringing this up. We did use scoop size in our calculations. Formula purchases were initially reported in total kg of powder purchased.  We used this, along with scoop size, to calculate the equivalent amount of prepared liquid formula this presented for each type of formula.  Then we calculated the weighted average iron and DHA concentration represented, controlling for the amount of each individual formula purchased.  We think this is a significant strength of the work and have added text to our methods to hopefully resolve our miscommunication.

The paragraph on formula staging regulations could be referenced (line 48 to 54).

Thank you.  We have added references to these sentences.